# Evaluation of the Synthesis and Skin Penetration Pathway of Folate-Conjugated Polymeric Micelles for the Dermal Delivery of Irinotecan and Alpha-Mangostin

**DOI:** 10.3390/pharmaceutics17081014

**Published:** 2025-08-05

**Authors:** Thanchanok Sirirak, Thirapit Subongkot

**Affiliations:** 1Department of Pharmacognosy and Pharmaceutical Chemistry, Faculty of Pharmaceutical Sciences, Burapha University, Chonburi 20131, Thailand; thanchanoks@go.buu.ac.th; 2Department of Pharmaceutical Technology, Faculty of Pharmaceutical Sciences, Burapha University, Chonburi 20131, Thailand

**Keywords:** Irinotecan hydrochloride, alpha-mangostin, folic acid, polymeric micelles, dermal delivery

## Abstract

**Background/Objectives**: The present study aimed to synthesize folate-conjugated poloxamers and develop polymeric micelles for the dermal delivery of irinotecan and alpha-mangostin for the treatment of melanoma using poloxamer 188 and poloxamer 184, which have never been synthesized with folate before. **Methods**: Poloxamer 188 and poloxamer 184 were synthesized with folate by esterification. The in vitro skin penetration enhancement of irinotecan- and alpha-mangostin-loaded folate-conjugated polymeric micelles was evaluated. The skin penetration pathway of folate-conjugated polymeric micelles was investigated by colocalization of multiple fluorescently labeled particles using confocal laser scanning microscopy (CLSM). **Results**: Folate-conjugated poloxamer 188 and poloxamer 184 were successfully synthesized. The prepared irinotecan- and alpha-mangostin-loaded folate-conjugated polymeric micelles from poloxamer 188 and poloxamer 184 had particle sizes of approximately 180 and 150 nm, respectively, indicating a positive charge with a narrow size distribution which could be easily taken up into cells. An in vitro skin penetration study revealed that folate-conjugated polymeric micelles from poloxamer 184 significantly enhanced the skin penetration of irinotecan and alpha-mangostin to a greater extent than the solution. CLSM visualization revealed that folate-conjugated polymeric micelles penetrated through the skin by the transfollicular pathway as the major penetration pathway, whereas penetration by the intercluster pathway, transcellular pathway and intercellular pathway constituted a minor pathway. **Conclusions**: Folate-conjugated poloxamer 184 polymeric micelles are promising candidates for the dermal delivery of anticancer drugs by the transfollicular pathway as the major skin penetration pathway.

## 1. Introduction

Skin cancer currently has a high incidence rate in both Caucasians and Asians [1,2,3]. Skin cancer can be classified as nonmelanoma or melanoma skin cancer. Nonmelanoma skin cancer originates from keratinocytes and can be divided into basal cell carcinoma and squamous cell carcinoma [4]. Melanoma skin cancer is caused by the abnormal proliferation of melanocytes in the epidermis. Melanocytes are located in the stratum basale and produce melanin pigments to protect the skin from ultraviolet radiation. Melanoma skin cancer patients have a higher mortality rate than nonmelanoma skin cancer patients because the cancer cells can spread to vital organs.

The treatments for skin cancer are surgical excision, radiation therapy, chemotherapy and targeted therapy [5,6]. Chemotherapy is widely used for cancer treatment because of its affordability. Conventional chemotherapy kills both cancerous and normal cells. The development of a targeted drug delivery system for cancer cells is necessary to avoid the side effects of chemotherapy. Folate receptors are overexpressed on the surface of various cancer cells, including kidney, lung, ovarian, endometrial, breast, bladder and pancreatic cancer cells [7]. Folate receptors are also expressed in melanoma cell lines and melanoma tissues [8]. Thus, the folate receptor, as a specific cellular marker, can be used for the development of targeted nanoparticles for the selective delivery of chemotherapeutic drugs to increase the efficacy of treatment and reduce the side effects of skin cancer treatment.

Alpha-mangostin is a natural xanthone extracted from the mangosteen pericarp [9]. Alpha-mangostin has been shown to have a chemotherapeutic effect by inhibiting the proliferation, migration and invasion of human breast cancer cells [10,11,12], lung cancer cells [13] and melanoma cells [14]. With regard to skin cancer treatment, mice treated with alpha-mangostin exhibit reduced tumor formation and growth due to the inhibition of proinflammatory factors and the promotion of anti-inflammatory factors in tumors [15]. Alpha-mangostin also inhibits angiogenesis, which is an essential process for tumor progression and metastasis [16]. Irinotecan, a camptothecin-derivative anticancer drug, has been used for the treatment of various malignancies because it inhibits the topoisomerase I enzyme [17].

A poloxamer is an amphiphilic nonionic triblock copolymer consisting of a central hydrophobic core of polyoxypropylene with two hydrophilic side chains of polyoxyethylene. Owing to their amphiphilic properties, poloxamers can arrange their structure to form polymeric micelles in water, with a hydrophobic core for lipophilic molecule entrapment. To deliver drugs specifically to cancer cells by utilizing the overexpression of folate (folic acid) receptors on the surface of many types of malignant cells, the present study synthesized poloxamers with folic acid. There were several studies synthesizing poloxamer 407 with folic acid for intravenous utilization [18,19]. The average molecular weight of poloxamer 407 is 12.6 KDaltons [20]. The average molecular weight of poloxamer 188 and poloxamer 184 is 7361 Daltons and 2900 Daltons, respectively [21,22]. To compare the physicochemical properties of folate-conjugated poloxamers having a molecular weight lower than poloxamer 407, in the present study, two types of poloxamers, namely, poloxamer 188 and poloxamer 184, were used to synthesize folate-conjugated polymeric micelles, and their in vitro skin penetration efficiencies were compared. Both folate-conjugated poloxamer 188 and folate-conjugated poloxamer 184 were loaded with irinotecan and alpha-mangostin for synergistic effects on the inhibition of cancer cell growth.

The stratum corneum, the outermost layer of the skin, plays an important role in skin absorption by limiting the skin penetration of drugs. Theoretically, molecules that can easily penetrate the skin should have a log partition coefficient (log P) between 1 and 3, with a molecular weight lower than 500 Daltons [23]. Irinotecan has a log P of 1.5 and a molecular weight of 623.1 Daltons [24]. Alpha-mangostin has a log P of 5.935 and a molecular weight of 410.46 Daltons [25]. Therefore, both irinotecan and alpha-mangostin exhibit poor skin penetration. The present study investigated the codelivery of irinotecan and alpha-mangostin using folate-conjugated polymeric micelles as a cancer-targeted drug delivery system. The skin penetration pathway of folate-conjugated polymeric micelles was also investigated.

## 2. Materials and Methods

### 2.1. Materials

Irinotecan hydrochloride trihydrate, alpha-mangostin, 1′-carbonyldiimidazole (CDI), folate (FA), Nile red and sodium 1-octanesulfonate were purchased from Tokyo Chemical Industry Co., Ltd., Tokyo, Japan. Poloxamer 188 (Kolliphor^®^ P 188) and poloxamer 184 (Pluracare L64G) were purchased from BASF, Ludwigshafen, Germany. 4′,6-Diamidino-2-phenylindole dihydrochloride (DAPI) was purchased from Sigma Aldrich, MO, USA. N-(7-Nitrobenz-2-Oxa-1,3-Diazol-4-yl)-1,2-dihexadecanoyl-sn-glycero-3-phosphoethanolamine, triethylammonium salt (NBD-PE) was purchased from Invitrogen, CA, USA. All other reagents were of analytical grade and were commercially available.

### 2.2. Synthesis of FA-Conjugated Poloxamers

The synthesis processes of FA-conjugated poloxamer 188 (FA-P188) and FA-conjugated poloxamer 184 (FA-P184) are shown in Figure 1 and Figure 2, respectively. The synthesis reaction was modified from the method of Butt et al. [19]. Briefly, FA was dissolved in DMSO for 8 h until it was completely dissolved. After adding CDI, the solution was stirred overnight under a nitrogen atmosphere and protected from light. P188 (or P184) was added to the solution, and the mixture was continuously stirred under similar conditions for 24 h. The mixture was dialyzed against ultrapure water for 5 days. The dialyzed solution was lyophilized to recover FA-P188 or FA-P184.

The conjugation of FA to each poloxamer was confirmed using ^1^H NMR (Bruker Avance III HD 400 spectrometer, Bruker Corporation, MA, USA) by dissolving FA, P188, P184, FA-P188 and FA-P184 in deuterated DMSO. The ^1^H NMR spectra of FA, P188, P184, FA-P188 and FA-P184 were analyzed to ensure the success of these synthesis processes.

### 2.3. Evaluation of the Critical Micelle Concentration (CMC) of the FA-Conjugated Poloxamers

The CMC is the minimum concentration of surfactant at which molecules can form micelles. The CMCs of the FA-conjugated poloxamers (FA-P188 and FA-P184) were determined by a fluorescence spectrometer (PerkinElmer LS55, PerkinElmer, Llantrisant, UK) using Nile red as a fluorescent probe [26]. A Nile red solution was prepared at a concentration of 100 µM by dissolving it in tetrahydrofuran. FA-conjugated poloxamer was accurately weighed into a volumetric flask before being dissolved in water to prepare a stock solution (2 mg/mL). Then, the stock solution was pipetted into a 5 mL volumetric flask to prepare various concentrations of FA-conjugated poloxamer ranging from 1 to 1000 µg/mL, and 50 µL of Nile red solution was added before volume adjustment. The final concentration of Nile red in different concentrations of the FA-conjugated poloxamer solution was 1 µM. The fluorescence intensity (FI) of the prepared solution was measured at an excitation wavelength of 550 nm and an emission wavelength of 633 nm. The average FI of each concentration was plotted against the log of the FA-conjugated poloxamer to determine the CMC value.

### 2.4. Formulation of Irinotecan- and Alpha-Mangostin-Loaded FA-Conjugated Polymeric Micelles

Irinotecan- and alpha-mangostin-loaded FA-conjugated polymeric micelles were prepared by the direct dissolution method. The solvents used for the dissolution of irinotecan, alpha-mangostin and FA-conjugated poloxamer (FA-P188 or FA-P184) were composed of ethanol–propylene glycol–water (EPW) at a ratio of 37.72:37.72:24.56 *w*/*w*/*w*. The formulations of the irinotecan and alpha-mangostin solution, as well as the irinotecan- and alpha-mangostin-loaded polymeric micelles, are shown in Table 1. For each FA-conjugated poloxamer, irinotecan and alpha-mangostin were accurately weighed into a 5 mL volumetric flask. Then, 4 mL of EPW was added to the volumetric flask. The volumetric flask was placed in a sonicator bath until the components were completely dissolved. EPW was added to adjust the volume to 5 mL. The irinotecan and alpha-mangostin solution was prepared as described above without the addition of the FA-conjugated poloxamer.

### 2.5. Characterization of Irinotecan- and Alpha-Mangostin-Loaded Polymeric Micelles

The particle size, polydispersity index (PDI) and surface charge (zeta potential) of the irinotecan- and alpha-mangostin-loaded FA-conjugated polymeric micelles were analyzed by dynamic light scattering using a particle size analyzer (Zetasizer Nano-ZS; Malvern Instrument, Worcestershire, UK). Prior to the measurement, each sample was diluted with an appropriate amount of ultrapure water at a ratio of 1:9 *v*/*v* before being placed into a folded capillary zeta cell, which was then closed with a stopper. Each measurement was analyzed in triplicate.

### 2.6. In Vitro Skin Penetration Study

#### 2.6.1. Skin Preparation

Pig abdominal skin was used as a barrier membrane for the skin penetration study. The abdominal skin was obtained from a local pig farm after the neonatal pig died naturally after birth. Burapha University did not require ethical approval for animal carcass usage. The abdominal skin was separated from the body, and the remaining muscle was carefully removed using a surgical blade. The obtained full-thickness skin had epidermal, dermal and subcutaneous tissues, with a skin thickness not greater than 1 mm. The skin was washed with Phosphate Buffer Saline (PBS) and placed on lint-free tissue paper (Kimberly-Clark Kimtech Science Kimwipes, Surrey, UK) before the skin penetration experiment.

#### 2.6.2. Skin Penetration Study

The following formulations were used for the penetration study: the irinotecan and alpha-mangostin solution (control); irinotecan- and alpha-mangostin-loaded FA-P188; and irinotecan- and alpha-mangostin-loaded FA-P184. The skin penetration study was performed by jacketed Franz diffusion cells connected with a water circulating bath to control the temperature at 32 °C throughout the experiment. Pig skin obtained as described in Section 2.6.1 was mounted between the donor and receiver compartments, with the stratum corneum facing the donor part. The receiver compartment, with a volume of approximately 6.5 mL, was filled with PBS as the receiver medium and stirred with a magnetic stirring bar, and 0.5 mL of each formulation was pipetted into the donor compartment and covered with Parafilm^®^. After treatment for 6 h, 1 mL of the receiver medium was withdrawn through the sampling port to analyze the amount of irinotecan and alpha-mangostin by HPLC. The remaining formulation in the donor compartment was removed, and the skin was washed with PBS three times. The amount of drugs that penetrated the skin was determined by the tape strip method as described by Subongkot and Sirirak [27]. The treated skin was cut to obtain only the penetrated area and fixed onto paraffin wax. The stratum corneum was then removed by stripping with 24 mm wide adhesive tape (Scotch^®^ Transparent Tape 600, 3M, Bangkok, Thailand) 35 times. All stripped tapes, except the first one, were immersed into a screw-capped glass vial containing 5 mL of methanol. The vial was placed in a sonicator bath for 15 min, and 1 mL of methanol was withdrawn for drug quantification by HPLC. The amount of irinotecan and alpha-mangostin that penetrated the stratum corneum was calculated by the following equation (Equation (1)):Drug amount in the stratum corneum (µg/cm^2^) = drugs in the stratum corneum (µg)/skin penetration area (cm^2^)(1)

After removing the stratum corneum, the drug amount was determined in the remaining viable epidermis and dermis, which were cut into small pieces and immersed in a screw-capped glass vial containing 3 mL of methanol. The vial was placed in a sonicator bath for 15 min, and 1 mL of methanol was withdrawn to measure the drug content by HPLC. The amount of irinotecan and alpha-mangostin that penetrated the viable epidermal and dermal tissues was calculated by the following equation (Equation (2)):Drug amount in the viable epidermis and dermis (µg/cm^2^) = drugs in the viable epidermis and dermis (µg)/skin penetration area (cm^2^)(2)

### 2.7. Skin Penetration Pathway Evaluation

Confocal laser scanning microscopy (CLSM) was utilized to investigate the skin penetration pathway of polymeric micelles by the colocalization technique using different fluorescent colors for the entrapped drugs and particles. Rhodamine B, a lipophilic red fluorescent compound with a log partition coefficient (log P) of 1.95 [28], which is similar to that of irinotecan (log P = 1.5) [24], was used as the entrapped drug. The structure of NBD-PE is a phosphatidylethanolamine probe with NBD, which results in green fluorescence. Therefore, NBD-PE is a fluorescent surfactant that can be used to label polymeric micelle particles. The formulation providing the highest penetration amount of irinotecan and alpha-mangostin in the viable epidermal and dermal tissues described in Section 2.6.2 was used for the skin penetration pathway evaluation.

#### 2.7.1. Preparation of Fluorescently Labeled Polymeric Micelles

FA-conjugated poloxamer, rhodamine B base and NBD-PE (10 mg, 8 mg and 3.2 mg, respectively) were weighed with an analytical balance into a 2 mL volumetric flask. The flask was filled with EPW, and the volumetric flask was placed in a sonicator bath until the components were completely dissolved. The EPW was subsequently adjusted to 2 mL.

#### 2.7.2. Skin Penetration Test

The penetration study was performed with Franz diffusion cells as described in Section 2.6.2. The fluorescently labeled polymeric micelles (400 µL) obtained from Section 2.7.1 were pipetted into the donor compartment of the diffusion cells without the addition of receiver medium. After 30 min, 1 h, 2 h and 6 h, the samples were withdrawn, and the skin was washed with PBS 3 times to remove the excess drug before being examined by CLSM.

#### 2.7.3. Preparation of Cross-Sectional Tissue

A portion of the treated skin obtained at 30 min, 1 h and 2 h from Section 2.7.2 was cross-sectioned at −30 °C using a cryomicrotome (Leica 1850, Leica Instruments GmbH, Nussloch, Germany). The tissue was embedded with cryostat embedding medium (Killik, Bio-Optica, Milano, Italy) and sectioned at a thickness of 5 µm onto positively charged microscope slides (Bio-Optica, Milano, Italy). DAPI was used to stain the nuclei of the cells to indicate viable epidermis and dermis. The sample was stained with 10 µg/mL DAPI solution for 15 s and washed by immersion in ultrapure water. The slide was allowed to dry at room temperature and immediately mounted with mounting medium (Bio Mount HM, Bio-Optica, Milano, Italy) before visualization by CLSM.

#### 2.7.4. CLSM Study

The treated skin and the sectioned samples were visualized using an inverted confocal laser scanning microscope (Zeiss LSM 800-Airy scan; Carl Zeiss, Jena, Germany). The skin was placed on a 22 × 50 mm glass square coverslip before being observed with a 10× objective lens. The red fluorescence of the rhodamine B base was detected at excitation and emission wavelengths of 577 and 603 nm, respectively. The green fluorescence of NBD-PE was detected at excitation and emission wavelengths of 493 and 517 nm, respectively. The microscope slides containing the sectioned specimens were visualized for red and green fluorescence as described above. The blue fluorescence of DAPI was detected at excitation and emission wavelengths of 353 and 465 nm, respectively. The images were obtained and analyzed with Zen software (Carl Zeiss Microscopy GmbH, Jena, Germany).

### 2.8. High-Performance Liquid Chromatography (HPLC) Analysis

Irinotecan and alpha-mangostin were analyzed using HPLC equipped with a photodiode array detector (Shimadzu, Kyoto, Japan). Irinotecan was analyzed using acetonitrile–methanol–phosphate buffer (pH 3.65 = 30:15:55 *v*/*v*/*v*) as the mobile phase. Phosphate buffer was prepared by dissolving 2.8 g of monobasic sodium phosphate monohydrate and 1.8 g of 1-octanesulfonic acid sodium in 1 L of water. Alpha-mangostin was analyzed using acetonitrile–methanol–phosphate buffer (pH 3.65 = 27:63:10 *v*/*v*/*v*) as the mobile phase. The analysis was performed using a C18 reversed-phase column (5 µm, 4.6 mm × 150 mm; InertSustain C18, GL Sciences, Tokyo, Japan). Each sample was injected with 15 µL of mobile phase at a flow rate of 1.5 mL/min. The column temperature was maintained at 40 °C, and the detector wavelength was 255 nm.

For the analysis of irinotecan, the method provided good linearity (R^2^ = 1) in the concentration range of 0.46875–15 µg/mL. The limit of detection and limit of quantitation were 0.097 and 0.294 µg/mL, respectively. The accuracy of the method was 99.61 ± 0.72%, and the precision was 0.85% [expressed as relative standard deviation (RSD)].

For the analysis of alpha-mangostin, the standard was prepared at concentrations ranging from 0.001563 to 1 µg/mL, which demonstrated good linearity (R^2^ = 0.9992). The limits of detection and quantification were 0.004184 µg/mL and 0.012677 µg/mL, respectively. The accuracy of the method was confirmed, with a recovery of 101.65 ± 0.75%. The precision was 0.74% (expressed as RSD).

### 2.9. Statistical Analysis

All the data were statistically analyzed using paired samples *t* tests, Student’s *t* tests and one-way analysis of variance (ANOVA) followed by a post hoc test (LSD). *p* < 0.05 was considered statistically significant.

## 3. Results and Discussion

### 3.1. Synthesis of the FA-Conjugated Poloxamer

#### 3.1.1. Synthesis and Characterization of FA-P188

Folate-conjugated P188 (FA-P188) was confirmed using ^1^H NMR (Figure 3). The ^1^H NMR spectrum of FA-P188 showed signals at 6.67 ppm and 7.64 ppm, which represented characteristic patterns of the aromatic protons of FA, as well as at 8.65 ppm, which corresponded to the H-7 position of FA. The peak at 1.05 ppm indicated the methyl group of P188, and the region of 3.34–3.69 ppm represented the -CH_2_CHO and -CH_2_CH_2_O- of the propylene oxide unit and the ethylene oxide unit of the poloxamer. The additional ester signal at 4.57 ppm, which corresponded to CH_2_ adjacent to the ester bond, confirmed ester conjugation between folate and P188.

#### 3.1.2. Synthesis and Characterization of FA-P184

Folate-conjugated P184 (FA-P184) was confirmed using ^1^H NMR (Figure 4). The ^1^H NMR spectrum of FA-P184 showed signals at 6.63 ppm and 7.65 ppm, which represented characteristic patterns of the aromatic protons of FA, as well as at 8.65 ppm, which corresponded to the H-7 position of FA. The peak at 1.05 ppm indicated the methyl group of P184, and the region of 3.35–3.67 ppm represented the -CH_2_CHO and -CH_2_CH_2_O- of the propylene oxide unit and the ethylene oxide unit of the poloxamer. The additional ester signal at 4.57 ppm, which corresponded to CH_2_ adjacent to the ester bond, confirmed ester conjugation between folate and P184.

### 3.2. Evaluation of the Critical Micelle Concentration (CMC) of the FA-Conjugated Poloxamer

The graphs of the FI and log concentration of FA-P188 and FA-P184 are shown in Figure 5a,b, respectively. According to the graphs, the CMCs of FA-P188 and FA-P184 were 426.58 and 501.19 µg/mL, respectively. In the present study, polymeric micelles were prepared from FA-P188 or FA-P184 at a concentration of 5 mg/mL, which was higher than the obtained CMCs.

### 3.3. Physicochemical Properties of FA-Conjugated Polymeric Micelles

The average particle size, polydispersity index (PDI) and zeta potential of polymeric micelles obtained from blank FA-conjugated polymeric micelles and drug (irinotecan and alpha-mangostin)-loaded FA-conjugated polymeric micelles are shown in Table 2. The mean particle size of drug-loaded FA-P188 was larger than that of blank FA-P188, whereas the mean particle size of drug-loaded FA-P184 did not significantly differ from that of blank FA-P184. These findings indicated that the addition of drugs increases the particle size only in FA-P188.

The mean particle sizes of drug-loaded FA-P188 and drug-loaded FA-P184 were 169.81 nm and 149.83 nm, respectively. The mean particle size of drug-loaded FA-P188 was significantly larger than that of drug-loaded FA-P184. The average PDIs of drug-loaded FA-P188 and drug-loaded FA-P184 were 0.1917 and 0.1048, respectively, indicating a narrow size distribution. The mean zeta potentials of blank FA-P188 and drug-loaded FA-P188 were −7.37 and 3.01 mV, respectively. The mean zeta potentials of blank FA-P184 and drug-loaded FA-P184 were −5.33 and 4.87 mV, respectively. Structurally, folate has dicarboxylic acid side chains. The synthesis between poloxamer and folate uses one carboxylic acid to create an ester bond, leaving one carboxylic acid. Thus, the blank folate-conjugated poloxamer exhibited a negative charge. Irinotecan HCl is a positively charged molecule, whereas alpha-mangostin is a nonionic charged molecule. Therefore, the positive charge of drug-loaded folate-conjugated polymeric micelles resulted from the addition of irinotecan HCl. The positively charged nanoparticles tended to uptake into cells more than neutral and negatively charged particles [29].

### 3.4. Skin Penetration Study

Melanoma is a type of cancer that originates from melanocytes and is present in the viable epidermis (stratum basale). Therefore, the tape strip technique was used to remove the stratum corneum to determine the amount of drug penetration into the viable epidermis and dermis.

The amounts of irinotecan and alpha-mangostin that penetrated the stratum corneum, deeper skin (viable epidermis and dermis) and receiver media from various formulations are shown in Table 3. The amount of penetrated irinotecan from FA-P184 was significantly greater than that from FA-P188 and the solution. The amount of penetrated irinotecan did not differ between FA-P188 and the solution. The amount of penetrated alpha-mangostin from FA-P184 was significantly greater than that from the solution. The amount of penetrated alpha-mangostin did not differ between FA-P184 and FA-P188 or between FA-P188 and the solution.

Irinotecan and alpha-mangostin entrapped in FA-P184 polymeric micelles penetrated the viable epidermis and dermis more effectively than FA-P188 polymeric micelles and the solution. Various studies have reported that a smaller particle size results in greater skin penetration [27,30,31,32]. With respect to the results of the particle size measurement from Section 3.3, the particle size of the FA-P188 polymeric micelles was significantly larger than that of the FA-P184 polymeric micelles. Consequently, the greater amounts of irinotecan and alpha-mangostin in the viable epidermis and dermis from the FA-P184 polymeric micelles resulted from the smaller size of the FA-P184 polymeric micelles compared with that of the FA-P188 polymeric micelles.

### 3.5. Skin Penetration Pathway Evaluation

According to the in vitro skin penetration study, the FA-P184 polymeric micelles had the greatest skin penetration of irinotecan and alpha-mangostin. Therefore, the FA-P184 polymeric micelle formulation was used as a candidate to investigate the skin penetration pathway of polymeric micelles. Rhodamine B base was used as the entrapped drug, whereas NBD-PE was used to label polymeric micelle particles; rhodamine B base and NBD-PE exhibited red and green fluorescence, respectively.

The stratum corneum, which acts as a rate-limiting barrier for percutaneous absorption, is composed of cornified cells embedded in intercellular lipid lamellae. Penetration through cornified cells is called the transcellular pathway, whereas penetration via intercellular lipids is the intercellular pathway. Additionally, skin penetration can occur via hair follicles, which are involved in the transfollicular pathway.

The skin samples treated with rhodamine B base-loaded NBD-PE-labeled polymeric micelles, as observed by CLSM, are shown in Figure 6. Strong red and green fluorescence was observed in hair follicles, indicating greater penetration of polymeric micelles through hair follicles than through nonfollicular regions. A top view of a three-dimensional overlay image of skin treated with rhodamine B base-loaded NBD-PE-labeled polymeric micelles, as observed by CLSM, is shown in Figure 7. Green fluorescence was clearly observed in hair follicles, indicating that the deposition and penetration of polymeric micelles via hair follicles were greater than those in the nonfollicular region.

Cross-sectional images of skin treated with rhodamine B base-loaded NBD-PE-labeled polymeric micelles observed by CLSM at 0.5 h, 1 h and 2 h are shown in Figure 8. At 0.5 h, the red fluorescence of rhodamine B base and the green fluorescence of NBD-PE, which represented the entrapped drug and polymeric micelle particles, respectively, were more intense in hair follicles than in the nonfollicular region (Figure 8(a1,a2)). At 1 h, the red fluorescence of the rhodamine B base was still more intense in the hair follicles than in the nonfollicular region (Figure 8(b1)). The green fluorescence of NBD-PE was more intense in hair follicles and viable epidermis than in the nonfollicular region (Figure 8(b2)). At 2 h, the red fluorescence of rhodamine B base was more intense in hair follicles more than in the nonfollicular region (Figure 8(c1)). The intensity of the green fluorescence of NBD-PE was strong in hair follicles and in the viable epidermis and dermis (Figure 8(c2)).

In addition to the intercellular pathway, transcellular pathway and transfollicular pathway, pig skin also has an intercluster pathway [33]. Clusters are groups of cornified cells surrounded by canyons and can be seen as shallow wrinkles on the skin surface from the stratum corneum to the stratum basale of the epidermis. Nanoparticles, such as liposomes [33] and microemulsions [34], can penetrate through the intercluster pathway. Figure 9 shows the hair follicle and intercluster regions of pig skin treated with rhodamine B base-loaded NBD-PE-labeled polymeric micelles, as observed by CLSM. The green fluorescence representing polymeric micelle particles accumulated in hair follicles and in the intercluster region. To explore the penetration of polymeric micelle particles by the intercluster region, the marked area of Figure 9 was scanned by a laser to obtain images in the intercluster region. The serial x-z plane images of polymeric micelle particles by the intercluster region and overlay images are shown in Figure 10a and Figure 10b, respectively. The polymeric micelle particles penetrated through the intercluster region up to 90 µm, as indicated by the green fluorescence. The green fluorescence intensity in the intercluster region was greater than that in the non-intercluster region. Thus, these findings suggested that the intercluster pathway also facilitates the penetration of nanoparticles into deeper layers of the epidermis.

According to the top-view images and cross-sectional images of pig skin treated with multifluorescent labeled particles as visualized by CLSM, the present findings revealed that polymeric micelles penetrate through the skin by the transfollicular pathway as a major skin penetration pathway, whereas penetration by the intercluster pathway, transcellular pathway and intercellular pathway constitute a minor skin penetration pathway. The transfollicular pathway plays an important role in enhancing the percutaneous absorption rate of drugs entrapped in nanoparticles by bypassing the stratum corneum to reach the viable epidermis and dermis for dermal and transdermal drug delivery of nanoparticles such as ultradeformable liposomes [35] and microemulsions [34]. The penetration through the intercellular pathway and transcellular pathway employs passive diffusion of drug. Therefore, the bypass of the stratum corneum barrier to viable epidermis and dermis provided an essential channel which could facilitate penetration of an amount of drug greater than the intercluster pathway, intercellular pathway and transcellular pathway.

## 4. Conclusions

The present study successfully synthesized folate-conjugated polymeric micelles using poloxamer 188 and poloxamer 184. The prepared polymeric micelles were nanoscale-sized particles with a positive charge. The in vitro skin penetration study revealed that the folate-conjugated polymeric micelles (FA-P184) significantly increased the skin penetration of irinotecan and alpha-mangostin compared with that of the control (solution). The in vivo anticancer efficacy of these compounds will be investigated in future studies. The polymeric micelle particles penetrated through the skin by the transfollicular pathway as the major penetration pathway, whereas the intercluster pathway, transcellular pathway and intercellular pathway were minor penetration pathways.

## Figures and Tables

**Figure 1 pharmaceutics-17-01014-f001:**
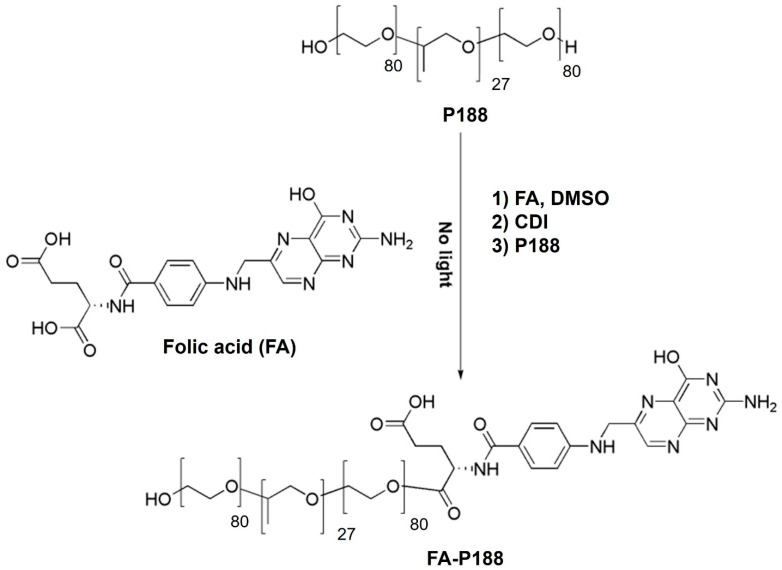
Synthesis of FA-conjugated poloxamer 188.

**Figure 2 pharmaceutics-17-01014-f002:**
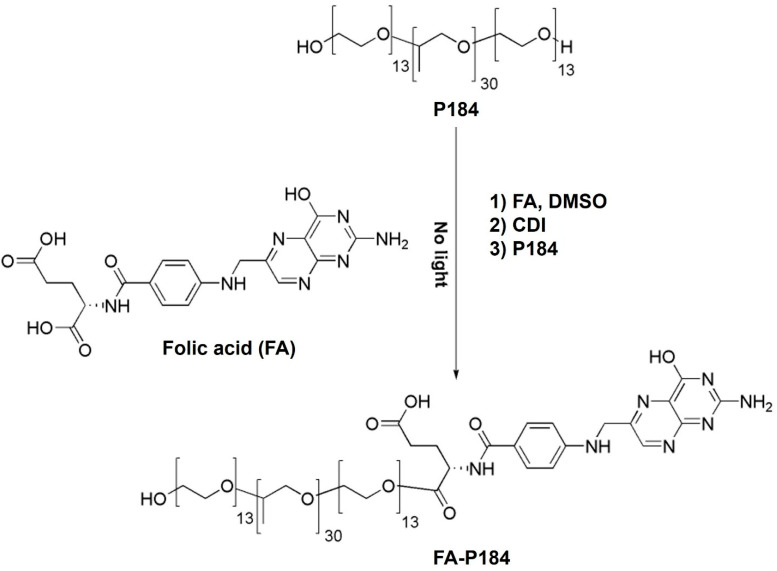
Synthesis of FA-conjugated poloxamer 184.

**Figure 3 pharmaceutics-17-01014-f003:**
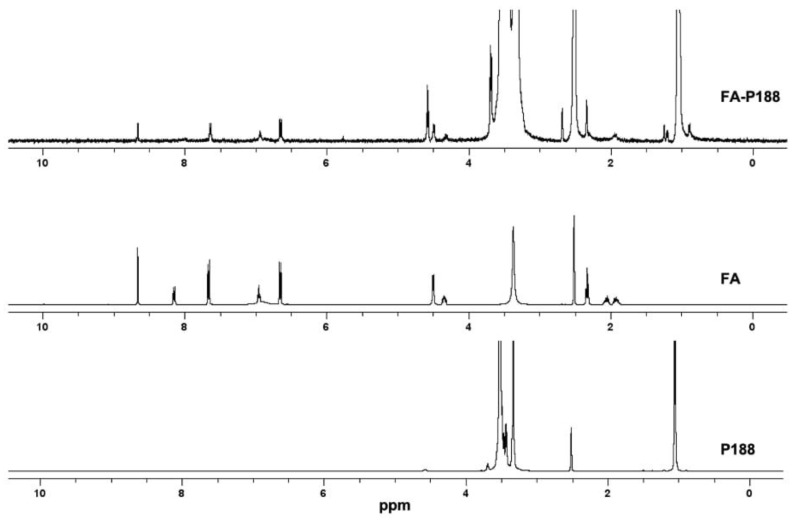
^1^H NMR spectra of FA-P188, FA and P188. Abbreviations: P188 = poloxamer 188, FA = folic acid.

**Figure 4 pharmaceutics-17-01014-f004:**
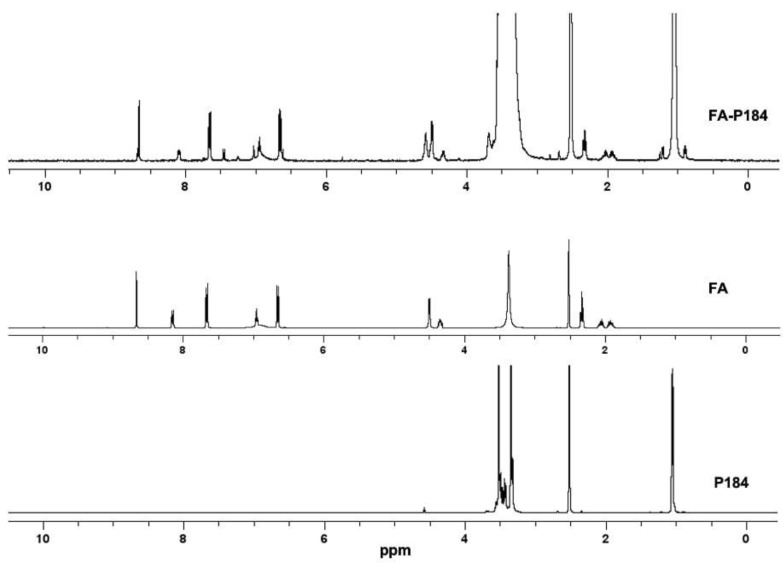
^1^H NMR spectra of FA-P184, FA and P184. Abbreviations: P184 = poloxamer 184, FA = folic acid.

**Figure 5 pharmaceutics-17-01014-f005:**
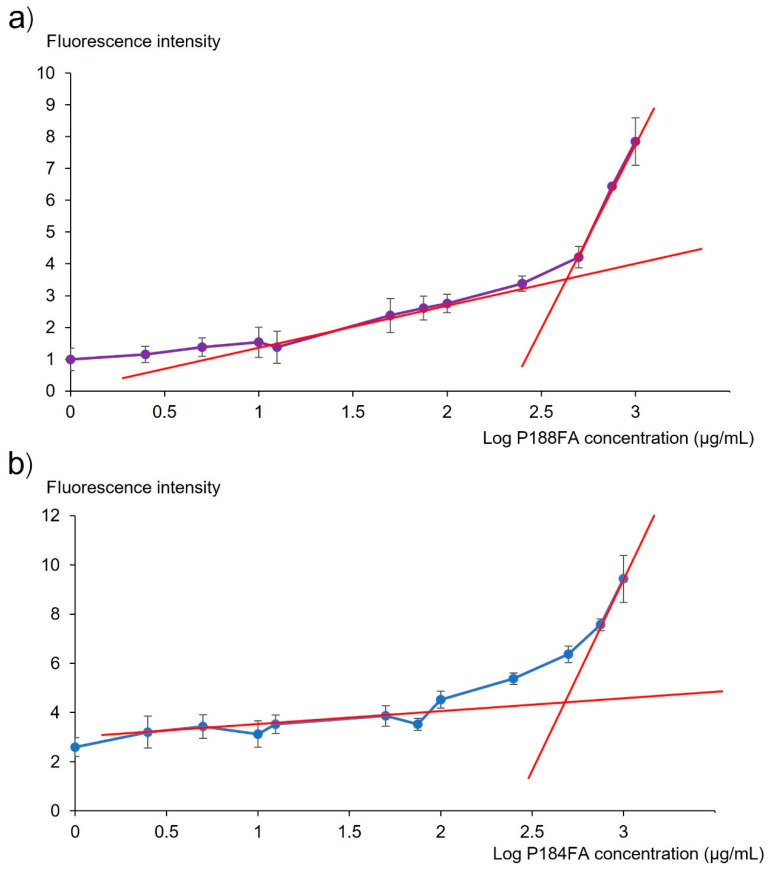
Critical micelle concentrations of (**a**) FA-P188 and (**b**) FA-184. Purple and blue dots represent concentration of P188FA and P184FA, respectively. The intersection between red lines indicate the CMC value.

**Figure 6 pharmaceutics-17-01014-f006:**
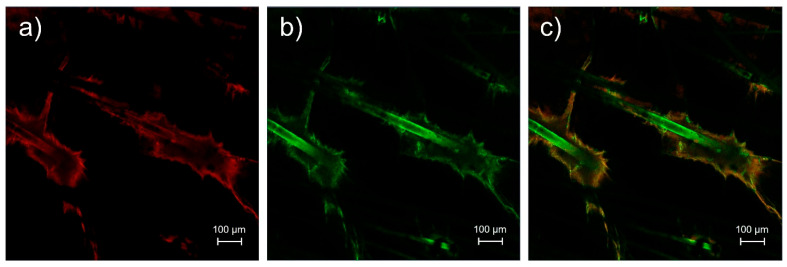
Top-view images of pig skin treated with rhodamine B base-loaded NBD-PE-labeled polymeric micelles at 6 h; (**a**) red fluorescence of the rhodamine B base, (**b**) green fluorescence of NBD-PE and (**c**) merged image of (**a**,**b**). The scale bar represents 100 µm.

**Figure 7 pharmaceutics-17-01014-f007:**
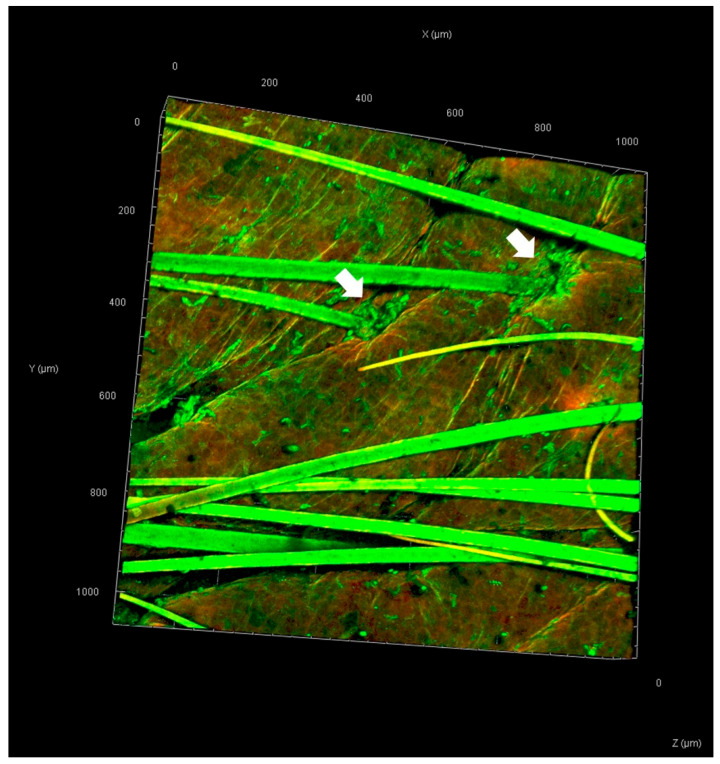
Top-view three-dimensional overlay image of pig skin treated with rhodamine B base-loaded NBD-PE-labeled polymeric micelles at 6 h. White arrows indicate the accumulation of polymeric micelle particles. The scale represents µm.

**Figure 8 pharmaceutics-17-01014-f008:**
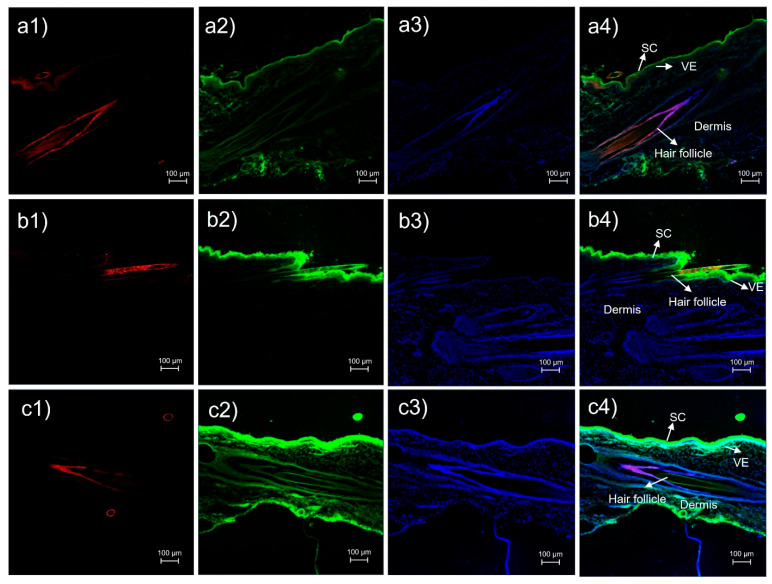
Cross-sectional images of pig skin treated with rhodamine B base-loaded NBD-PE-labeled polymeric micelles stained with DAPI at (**a**) 0.5 h, (**b**) 1 h and (**c**) 2 h; (**1**) red fluorescence of the rhodamine B base, (**2**) green fluorescence of NBD-PE, (**3**) blue fluorescence of DAPI and (**4**) merged image of (**1**), (**2**) and (**3**). The scale bar represents 100 µm. SC = stratum corneum and VE = viable epidermis.

**Figure 9 pharmaceutics-17-01014-f009:**
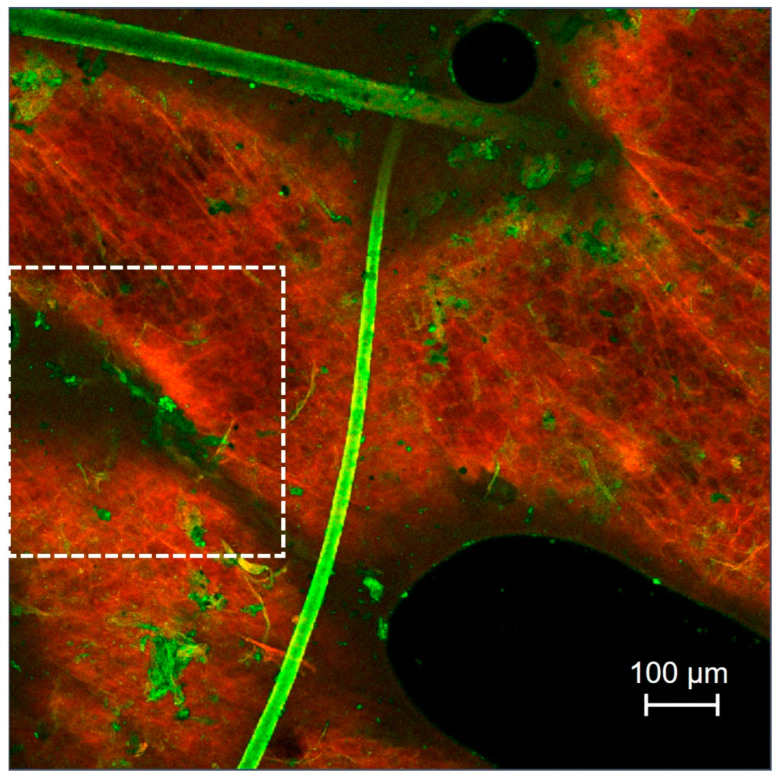
Top-view images of the intercluster region (white dashed line square) of pig skin treated with rhodamine B base-loaded NBD-PE-labeled polymeric micelles. The scale bar represents 100 µm.

**Figure 10 pharmaceutics-17-01014-f010:**
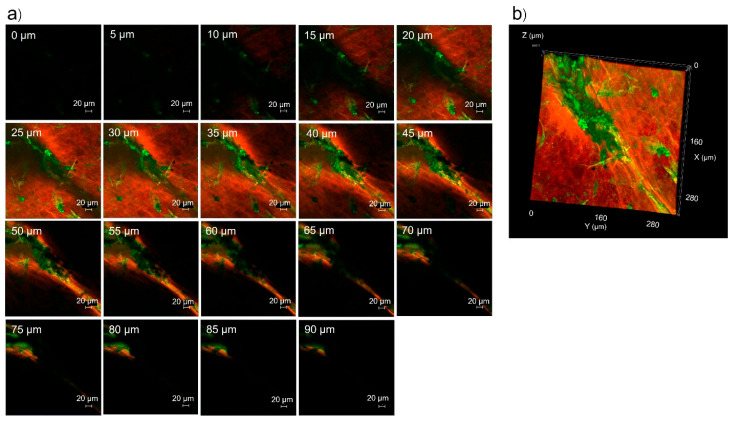
(**a**) Serial x-z plane images of the marked area from Figure 9, (**b**) overlay images of (**a**).

**Table 1 pharmaceutics-17-01014-t001:** Components of the solution and polymeric micelle formulation.

Formulation	Folate-Conjugated Polymer (mg)	Irinotecan (mg)	Alpha-Mangostin (mg)	EPW qs
Solution	-	15	5	5 mL
FA-P188	25	15	5	5 mL
FA-P184	25	15	5	5 mL

EPW = ethanol–propylene glycol–water = 37.72:37.72:24.56 *w*/*w*/*w*.

**Table 2 pharmaceutics-17-01014-t002:** Physicochemical parameters of blank polymeric micelles and drug (irinotecan and alpha-mangostin)-loaded polymeric micelles.

Formulation	Particle Size (nm)	PDI	Zeta Potential (mV)
Blank FA-P188	142.97 ± 1.72	0.3813 ± 0.0479	−7.37 ± 1.44
Drug-loaded FA-P188	169.81 ± 3.74	0.1917 ± 0.0156	3.01 ± 1.74
Blank FA-P184	152.37 ± 3.67	0.4367 ± 0.0484	−5.33 ± 1.11
Drug-loaded FA-P184	149.83 ± 3.19	0.1048 ± 0.0108	4.87 ± 0.24

Each value represents the mean ± standard deviation (*n* = 3).

**Table 3 pharmaceutics-17-01014-t003:** Skin penetration results of irinotecan and alpha-mangostin from each formulation.

Formulation	Stratum Corneum		Deeper Skin		Receiver Medium	
	Irinotecan (µg/cm^2^)	Alpha-Mangostin (µg/cm^2^)	Irinotecan (µg/cm^2^)	Alpha-Mangostin (µg/cm^2^)	Irinotecan (µg/cm^2^)	Alpha-Mangostin (µg/cm^2^)
Solution	16.05 ± 4.88	6.49 ± 2.81	2.68 ± 0.58	0.65 ± 0.11	ND	ND
FA-P188	61.21 ± 98.77	16.68 ± 20.74	2.14 ± 1.58	1.61 ± 1.04	ND	ND
FA-P184	20.05 ± 19.81	3.18 ± 1.69	8.73 ± 1.34	2.85 ± 1.39	ND	ND

Each value represents the mean ± standard deviation (*n* = 3). Deeper skin consists of the viable epidermis and dermis. ND = not detected.

## Data Availability

The data that support the findings of this study are available from the corresponding author upon reasonable request.

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
