# Peer review of "Evaluation of the Synthesis and Skin Penetration Pathway of Folate-Conjugated Polymeric Micelles for the Dermal Delivery of Irinotecan and Alpha-Mangostin"

_pharmaceutics, 2025, doi:10.3390/pharmaceutics17081014_

Round 1

Reviewer 1 Report

Comments and Suggestions for Authors

This manuscript presents a well-designed and innovative study with clear implications for targeted dermal drug delivery in melanoma therapy. With major revisions to address the given below points, it is suitable for publication.

  1. The authors should ensure all abbreviations are defined at first use.
  2. Language and formatting errors are present; a careful proofreading would improve readability.
  3. Data on the long-term stability of the micelles under storage conditions would be valuable for translational relevance.
  4. While the transfollicular pathway is highlighted, further mechanistic studies could provide deeper understanding.
  5. Statistical methods are mentioned, but more detail on sample size justification and power analysis would strengthen the rigor.
  6. The study is limited to in vitro experiments, thus suggested to indicate correlation via previous published similar work on animals or future work should include in vivo efficacy and safety studies to validate the clinical potential.
  7. The axis should be presented properly, such as figure 5.
  8. If possible improve the scale bar visibility  for figure 10. 
Comments on the Quality of English Language

Minor language and formatting errors are present; a careful proofreading would improve readability.

Author Response

This manuscript presents a well-designed and innovative study with clear implications for targeted dermal drug delivery in melanoma therapy. With major revisions to address the given below points, it is suitable for publication.

  1. The authors should ensure all abbreviations are defined at first use.

We have tried to check that all abbreviations used in this manuscript are defined at first usage. 

  1. Language and formatting errors are present; a careful proofreading would improve readability.

We apologized for any error occurring in the manuscript. The error parts which we knew were edited. However, the English of this manuscript was already checked and edited by American Journal Expert.

  1. Data on the long-term stability of the micelles under storage conditions would be valuable for translational relevance.

Thank you very much for your valuable suggestion. We will perform it in future study.

  1. While the transfollicular pathway is highlighted, further mechanistic studies could provide deeper understanding.

Thank you for your suggestion. The penetration through intercellular pathway and transcellular pathway employs passive diffusion of drug. Therefore, the bypass of stratum corneum barrier to viable epidermis and dermis provided an essential channel which could facilitate penetration amount of drug greater than intercluster pathway, intercellular pathway and transcellular pathway. This was mentioned in line 1-4 of page 21.

  1. Statistical methods are mentioned, but more detail on sample size justification and power analysis would strengthen the rigor.

 We appreciate the reviewer’s thoughtful comment. The statistical analysis used in this study was appropriate for scientific research. In our study, the triplicate measurements were performed as technical replicates and repeated measurements of the same sample were done to ensure analytical precision and reproducibility.

  1. The study is limited to in vitro experiments, thus suggested to indicate correlation via previous published similar work on animals or future work should include in vivo efficacy and safety studies to validate the clinical potential.

Thank you very much for your opinion. There have been no report using folate conjugated poloxamer for dermal drug delivery elsewhere. We already mentioned that the in vivo study in animal model will be performed for clinical application in next study in line 7-8 of page 21.

  1. The axis should be presented properly, such as figure 5.

Thank you very much. The Y axis of Figure 5 was edited.

  1. If possible improve the scale bar visibility for figure 10. 

Thank you very much. The scale bar of Figure 10 was edited for clear visualization.

Reviewer 2 Report

Comments and Suggestions for Authors

The manuscript investigates the penetration-enhancing formulations of anti-melanoma drugs through experiments. The study is well structured, the experiments are properly designed, and the results are transparent. However, some minor changes are needed before publication.

The role of the abstract is not only an extract of the article, but also to highlight the main points of the study. It should be rewritten accordingly.

In the Formulation of conjugated poloxamer section, the proportions of the solvent mixture used are quite strange. What justified this composition?

In section 3.3, the method of measuring the physicochemical properties is incompletely described, this needs to be supplemented. What does the appropriate amount of ultrapure water mean? What can be determined about the stability of the formulations, etc. from the zeta potential values?

In section 3.4. the amount of absorbed active ingredients in percentage comparison with the starting amount of active ingredient would be a useful and illustrative addition to the study.

Line 391 - does human skin also have a similar intercluster pathway? Is dead pig skin a suitable model for this type of absorption experiments?

Author Response

The manuscript investigates the penetration-enhancing formulations of anti-melanoma drugs through experiments. The study is well structured, the experiments are properly designed, and the results are transparent. However, some minor changes are needed before publication.

The role of the abstract is not only an extract of the article, but also to highlight the main points of the study. It should be rewritten accordingly.

  1. In the Formulation of conjugated poloxamer section, the proportions of the solvent mixture used are quite strange. What justified this composition?

The composition of mixture solvent used in this study resulted from the solubility study of our preliminary study which could provide the solubility of irinotecan and alpha-mangostin higher than using only water.

  1. In section 3.3, the method of measuring the physicochemical properties is incompletely described, this needs to be supplemented. What does the appropriate amount of ultrapure water mean? What can be determined about the stability of the formulations, etc. from the zeta potential values?

To determine the particle size, the sample must be diluted with an appropriate amount of water. In this study, we diluted the sample in water at a ratio of 1:9 v/v. We added this detail in section 2.5. For zeta potential and stability of nanoparticles, high zeta potential values (> ±30 mV) can stabilize nanoformulations via electrostatic repulsion.

However, the zeta potential of these polymeric micelle nanoparticles had positive charge. The positively charged nanoparticles tend to uptake into cells more than neutral and negatively charged particles. We added this issue in section 3.3.

  1. In section 3.4. the amount of absorbed active ingredients in percentage comparison with the starting amount of active ingredient would be a useful and illustrative addition to the study.

Thank you very much for your suggestion. We will include this type of data presentation in next study.

  1. Line 391 - does human skin also have a similar intercluster pathway? Is dead pig skin a suitable model for this type of absorption experiments?

Thank you very much for your suggestion. There has been no report about the intercluster pathway in human skin. However, this study found that intercluster pathway was a minor pathway for penetration study of nanoparticles. Therefore, the use of pig skin is acceptable for the experiment.

Reviewer 3 Report

Comments and Suggestions for Authors

The current manuscript deals with the design of micelles for precision therapy towards melanoma cancer. To this end, two different types of poloxamer were functionalized by folic acid and two anti-cancer drugs were loaded. the ex-vivo experiements were carried out using skin from neonatal pigs. 

In my opinion the Authors should deeply revise the manuscript since the rational of many choices is missing. As an example, it shuould be explained the selection of polymers, not only the drug. The new materials should be characterized also in terms of toxicity. Moreover, it is not clear the advantages of these new polymers with respect to the original one since there was not a control for comparison purpose. This is particularly important in case of ex-vivo studies: the aim is to have a target, but it was used skin from neonatal pig which is not a gold standard as model. 

Author Response

The current manuscript deals with the design of micelles for precision therapy towards melanoma cancer. To this end, two different types of poloxamer were functionalized by folic acid and two anti-cancer drugs were loaded. the ex-vivo experiements were carried out using skin from neonatal pigs.

In my opinion the Authors should deeply revise the manuscript since the rational of many choices is missing. As an example, it shuould be explained the selection of polymers, not only the drug. The new materials should be characterized also in terms of toxicity. Moreover, it is not clear the advantages of these new polymers with respect to the original one since there was not a control for comparison purpose. This is particularly important in case of ex-vivo studies: the aim is to have a target, but it was used skin from neonatal pig which is not a gold standard as model.

Thank you for your comments. The criteria for selection of polymer were polymer having molecular weight lower than poloxamer 407 and have not been reported by synthesizing with folic acid before. We added this explanation in line 14-19 of page 5.

This research used poloxamer conjugated with folic acid for topical usages. Poloxamer 188 and poloxamer 184 have been approved to use in cosmetic and skin care products as non-ionic surfactant. They have enough safety which is suitable to use regarding the objective of this study.

Because there have been no publications using folate conjugated poloxamer for topical dosage form application for cancer treatment. The neonatal pig skin usage was essential and appropriate to select a good candidate formulation for in vivo animal model in next experiment.

Round 2

Reviewer 1 Report

Comments and Suggestions for Authors

If possible reduce the similarity or else the authors have reflected all the said suggestions and comments, which made the manuscript enhanced with improved readability; Thus, I suggest for further consideration with acceptance.

Author Response

The manuscript was rewritten to reduce the similarity as seen in the attached file. 

Reviewer 2 Report

Comments and Suggestions for Authors

I accept the answers and revisions of the authors, and suggest the publication of the study.

Author Response

Thank you very much for your comments

Reviewer 3 Report

Comments and Suggestions for Authors

The Authors replied to the queries arisen by the Reviewer. 

Author Response

Thank you very much for your comments.